# Endometriosis, Pain, and Related Psychological Disorders: Unveiling the Interplay among the Microbiome, Inflammation, and Oxidative Stress as a Common Thread

**DOI:** 10.3390/ijms25126473

**Published:** 2024-06-12

**Authors:** Francesca Cuffaro, Edda Russo, Amedeo Amedei

**Affiliations:** 1Division of Interdisciplinary Internal Medicine, Careggi University Hospital of Florence, 50134 Florence, Italy; cuffarof@aou-careggi.toscana.it; 2Department of Clinical and Experimental Medicine, University of Florence, 50134 Florence, Italy; 3Network of Immunity in Infection, Malignancy and Autoimmunity (NIIMA), Universal Scientific Education and Research Network (USERN), 50139 Florence, Italy

**Keywords:** endometriosis, psychological disorders, microbiome, chronic pelvic pain, inflammation, oxidative stress, diet, nutrition

## Abstract

Endometriosis (EM), a chronic condition in endometrial tissue outside the uterus, affects around 10% of reproductive-age women, significantly affecting fertility. Its prevalence remains elusive due to the surgical confirmation needed for diagnosis. Manifesting with a range of symptoms, including dysmenorrhea, dyschezia, dysuria, dyspareunia, fatigue, and gastrointestinal discomfort, EM significantly impairs quality of life due to severe chronic pelvic pain (CPP). Psychological manifestations, notably depression and anxiety, frequently accompany the physical symptoms, with CPP serving as a key mediator. Pain stems from endometrial lesions, involving oxidative stress, neuroinflammation, angiogenesis, and sensitization processes. Microbial dysbiosis appears to be crucial in the inflammatory mechanisms underlying EM and associated CPP, as well as psychological symptoms. In this scenario, dietary interventions and nutritional supplements could help manage EM symptoms by targeting inflammation, oxidative stress, and the microbiome. Our manuscript starts by delving into the complex relationship between EM pain and psychological comorbidities. It subsequently addresses the emerging roles of the microbiome, inflammation, and oxidative stress as common links among these abovementioned conditions. Furthermore, the review explores how dietary and nutritional interventions may influence the composition and function of the microbiome, reduce inflammation and oxidative stress, alleviate pain, and potentially affect EM-associated psychological disorders.

## 1. Introduction

Endometriosis (EM) is a chronic, inflammatory disease characterized by the presence of ectopic endometrial tissue outside the uterine cavity [1,2]. This tissue comprises nerves, blood vessels, macrophages, and glandular and stromal cells [3]. Most cases of ectopic endometrium are found in the ovaries, but it can infiltrate different areas such as the pelvic peritoneum, uterosacral ligaments, fallopian tubes, and wide ligaments. Ectopic endometrium may also be observed in atypical locations both within and beyond the pelvic region, although such occurrences are less common [4]. Endometriosis is a prevalent gynaecological condition, affecting approximately 10% of women in their reproductive years [5]. Furthermore, EM stands as a primary contributor to female infertility [1,2]. Notably, 25 to 50% of women facing infertility are diagnosed with endometriosis, and additionally, 30 to 50% of EM patients encounter challenges in achieving pregnancy [6].

Moreover, EM is associated with heterogeneous symptoms such as chronic pelvic pain (CPP), defined as persistent pain perceived in the pelvic areas that occur over a period of 6 months [7]. It constitutes a substantial incapacitating factor, influencing 25% of menstruating women worldwide [7]. CPP is manifested as dysmenorrhea, dyschezia, dysuria, dyspareunia, and acyclic pelvic pain, but also as fatigue, gastrointestinal symptoms, and somatosensory amplification [1,2]. Notably, this symptom arises from the activation of macrophages and mast cells, contributing to an ongoing cycle of inflammation, oxidative stress, and pain [8,9,10]. While pain originates from endometrial lesions, its manifestation is often attributed to processes such as neuroinflammation, angiogenesis, and sensitization [11].

Given that an accurate diagnosis requires surgical visualization, the true prevalence of EM remains uncertain [2]. Modern imaging methods and related serum biomarkers have been proposed and may hasten the EM diagnosis, but surgical viewing, ideally laparoscopy with histological verification, is the main approach used to detect it [2,12]. Conventional treatments consist of surgical removal of endometriotic lesions, followed by hormonal therapy, which frequently has negative side effects and limited outcomes. In the absence of prolonged treatments, 50% of women undergoing surgery may require another procedure within 5 years. This recurrence can contribute to organ deterioration, exacerbated by function loss [13]. Remarkably, women with EM have been noted to show an increased likelihood of developing cardiovascular disease, rheumatoid arthritis, asthma, melanoma, ovarian, and breast cancer, attributed to the persistent inflammation and immunological dysregulation associated with this disease [14]. Moreover, due to the presence of CPP, endometriosis causes a decreased quality of life for women in their reproductive years [15,16]. Disturbances in the work environment, family connections, social life, self-esteem, mood symptoms, depression, and anxiety can also be experienced by EM patients [17,18,19]. Regarding associated psychological symptoms, CPP appears to be the primary mediator [20,21]. Hence, the main objectives of current treatments are to preserve fertility and provide sustained relief from symptoms [22].

Overall, endometriosis is the result of interconnected endocrine, inflammatory, immune, oxidative, and proangiogenic mechanisms, but its pathogenesis is still unclear. Recent hypotheses for EM genesis include Müllerianosis, coelomic metaplasia, extrauterine stem cell differentiation, benign lymphatic or hematogenous metastasis, bacterial contamination, and the most plausible theory, Sampson’s retrograde menstruation [23]. The latter refers to the reflux of menstrual fluid and endometrial cells into the peritoneal cavity through the fallopian tubes [24].

The latest research suggests that the microbiome imbalance could also contribute to the pathogenesis of endometriosis and associated CPP. Indeed, microbial dysbiosis in both the gut and reproductive tract can boost immune system dysregulation, induce elevated proinflammatory cytokines, and disrupt oestrogen metabolism. In addition, oxidative stress can instigate a comprehensive inflammatory response within the peritoneal cavity [25] that correlates to EM severity [26]. Both microbial dysbiosis and oxidative stress [27], when involved in the regulation of inflammation that gives rise to CPP, contribute to the development of EM-associated gastrointestinal symptoms and psychological comorbidities [28].

Therefore, the administration of diet and nutritional supplements in treating endometriosis and its associated symptoms encompasses the modulation of the gut microbiome, inflammation, oxidative stress, and oestrogen activity.

Within this framework, the main goal of this review is to dissect the relationships among endometriosis, pain, and associated psychological comorbidities. It aims to reveal the emerging role of the interplay among the microbiome, inflammation, and oxidative stress as a common thread. Furthermore, for the first time, the review will highlight the impact of dietary and nutritional interventions in regulating the above-mentioned interplay to relieve EM-associated pain and symptoms.

As a data source, we used the PubMed, Google Scholar, and Scopus databases for indexed English-language published material, and we examined reviews and original research articles, using combinations of keywords specific to each chapter. In detail, we used endometriosis, chronic pelvic pain, microbiome, oxidative stress, mood disorders, depression, anxiety, gut–brain axis, inflammation, diet, and nutrition. We analysed in vitro studies, animal models, observational clinical studies, and trials. We used our judgment to select articles and evidence and to interpret results. We considered articles and reviews published from 1994 to now.

## 2. Psychological Disorders Associated with Endometriosis

Although there are many symptoms reported by women with endometriosis, the most complex ones are frequently linked to psychiatric comorbidities and distress related to mental health, such as depression, anxiety, and increased stress, as well as poor sexual health, inadequate sleep quality, and generally low quality of life [15,29,30]. Endometriosis patients exhibited higher rates of any form of depression (18.9 compared to 9.3%) and anxiety (29.7 against 7.0%) than healthy controls, with a diminished quality of life, especially those who reported severe and painful symptoms [20,30,31,32,33,34,35,36,37].

In addition, the patient’s experience of endometriosis and related mental health symptoms is usually exacerbated by consequential psychosocial challenges and by CPP presence [37]. Notably, pain seems to be the main mediator between endometriosis and psychological conditions [20,21]. Indeed, the presence of psychiatric disorders is more strongly correlated with the pain intensity with respect to other aspects of the illness [33]. According to two recent meta-analyses, endometriosis and the associated CPP have a detrimental effect on mental health and quality of life [38,39]. Gambadauro et al. found that women with EM pain presented higher depressive symptoms compared to women without EM pain [21]. Furthermore, there is no significant difference between the depressive symptoms experienced by women without EM pain and those of healthy women [21]. Moreover, individual psychological factors are likely to influence how each patient experiences the EM-associated CPP. Regardless of pelvic pain, the psychological well-being of endometriosis patients is connected to their sense of self-esteem and self-efficacy [40]. Indeed, catastrophizing and negative pain-related thoughts are prevalent in women with symptomatic endometriosis, and these characteristics independently affect the quality of life [41,42].

In addition, a fascinating study focused on the phenotypic and genetic connections underlying the psychiatric comorbidities of endometriosis revealed that eating disorders, depression, and anxiety remained linked to endometriosis even after considering various concurrent conditions, such as CPP [43]. These phenotypic associations mirrored the genetic correlation observed between endometriosis and depression, anxiety, and eating disorders (EDs).

On the other hand, concerning EDs, there is limited available evidence [44]. An interesting study revealed that among endometriosis patients, pain is correlated with compromised eating behaviours. Significant differences were observed between women experiencing no or mild pain and those with moderate to severe pain regarding their attitudes, emotions, thoughts, and actions related to eating. Furthermore, Aupetit et al. investigated the correlation between EDs, irritable bowel syndrome (IBS), and endometriosis. Women with both conditions exhibited significantly higher anxiety and depression scores. These findings suggest a noteworthy association between IBS, ED, and endometriosis [45]. Another potential risk factor for ED emergence among women with endometriosis could be the high incidence of body image disturbances [46]. Indeed, a study reported that 77.3% of their EM cohort experienced disruptions in body image. However, the authors did not explore whether there was an association with EDs [46].

Additionally, treatment options for pain caused by endometriosis (such as ovarian suppression or advanced surgery) and infertility (such as assisted reproduction) sometimes yield unsatisfactory outcomes and are consistently convoyed by psychological discomfort. Unsurprisingly, poor access to care and interactions with caregivers are associated with lower psychological health in EM patients, potentially acting as moderators of mood symptoms [47]. Furthermore, endometriosis patients may encounter varying and diverse levels of health care, which can also influence their individual experiences. Notably, detection of and effective care for endometriosis are consistently hindered by a lack of non-invasive tests and limited knowledge among nonspecialized practitioners. Finally, the social stigma around endometriosis has a role in patients’ poor psychosocial well-being and delayed diagnosis [48].

## 3. Key Factors Influencing Chronic Pelvic Pain in Endometriosis

It has been reported that almost 50% of EM patients suffer from CPP [49] due to heterogenous pelvic pain, including dysmenorrhea, dyspareunia, non-menstrual (chronic) pelvic pain, discomfort during ovulation, dyschezia, and dysuria [50]. The pain seems to be independent of the EM stage, and so women with moderate disease may experience intense pelvic pain, while those with more severe EM may encounter less acute or chronic pain [51,52,53]. Patients mainly experience pain in the pelvis and abdomen, but also in the back and legs. The endometriotic lesions themselves can cause pain by offering afferent access to peripheral and central pain and sensory pathways. This is due to the vascularization and innervation of these lesions by sensory and autonomic fibres. However, lesions cannot be the exclusive source of pain [54]. Interestingly, given the abovementioned connection between endometriosis and psychiatric comorbidities, it seems that these comorbidities arise from the sensation of pelvic pain and its neurogenic origin, rather than being a direct consequence of endometriosis itself [20,55]. Nevertheless, it remains unclear whether psychological discomfort influences the pain perception or if psychological distress and psychopathological symptoms are consequences of the pain [30,56,57]. In addition, depression and anxiety may intensify the emotional and cognitive aspects of pain perception, leading to decreased pain tolerance and increased sensitivity to all physical symptoms, creating a vicious cycle [56]. Hence, distinguishing between endometriosis itself and pain is challenging, since they interact with each other through the same neurological, inflammatory, and immune-oxidative pathways [58,59].

In this scenario, the development of EM-associated pain can be attributed to a combination of underlying mechanisms (Figure 1), including inflammation, oxidative stress, neurogenic inflammation, peripheral and central sensitization, and cross-organ sensitization [11].

### 3.1. Inflammation

During menstruation, there is a natural decline in oestrogen and progesterone levels in the uterus. As a result, the endometrial tissue breaks down and is removed, serving to clear the uterus of menstrual debris and prepare it for a new cycle of endometrial regeneration. This process, orchestrated by the innate immune system, involves the recruitment of various immune cells, including neutrophils, macrophages, and natural killer cells (NKs), to assist in the breakdown of menstrual tissue. So, after going through apoptosis and necrosis, the uterine lining is eventually lost [60]. Alongside various other biological factors, programmed cell death during menstruation leads to the release of iron, ROS, prostaglandins, and a family of damage-associated molecular patterns (DAMPs) [61]. When retrograde menstruation takes place, endometrial fragments give rise to lesions in the peritoneum. These lesions, which retain their endometrial characteristics, express estrogenic receptors and undergo menstrual and immune-mediated events in a cyclical manner [62].

It has been reported that EM women’s peritoneal cavities include higher levels of inflammatory cytokines/chemokines, growth factors, neutrophils, and prostaglandins, which are visible at lesion sites [61]. In addition, the peritoneal fluid of EM patients with CPP is characterized by an elevated level of PGE2, tumour necrosis factor-α (TNF-α), nerve growth factor (NGF), and CCL5 (C-C chemokine ligand 5:), as well as interleukin (IL) IL-8 and IL-1β [63,64,65]. Crucially, all these inflammatory mediators can directly activate sensory nerve endings, inducing CPP [66,67,68]. Moreover, long-term exposure to proinflammatory cytokines is thought to activate and sensitize sensory nerves in endometriotic lesions, causing discomfort to be transferred to the central nervous system (CNS) [61]. Notably, the pain transition to the CNS is a critical step in the pain-processing pathway that contributes to different types of persistent visceral pain [69,70,71,72].

### 3.2. Oxidative Stress

As mentioned earlier, oxidative stress is believed to be a significant contributor to the inflammatory process in endometriosis [25]. ROS act as intermediates generated during oxygen metabolism; cells utilize antioxidant systems as a protective mechanism to counteract the ROS effects. However, an imbalance between ROS production and the availability of antioxidants, resulting in an excess of ROS and a deficiency in antioxidants, can lead to oxidative stress [73]. Specifically, superoxide and hydrogen peroxide (H_2_O_2_) are identified as ROS that play a role in regulating cellular proliferation within endometriosis by establishing connections with inflammation and extracellular matrix (ECM) degradation [74,75,76,77]. Excessive oxidative stress activates NF-κB, stimulating cellular processes that elevate cytokine production and trigger T helper Th1 and Th2 immune responses in pelvic endometriosis. [78]. In particular, IL-10 plays a crucial role in endometriosis development, as its expression triggers the activation of MMPs, ECM remodelling, and angiogenesis in serum and peritoneal fluid [78]. The IL-10 increase may be linked to the upregulated activation of the NF-κB signalling pathway resulting from oxidative stress and iron overload in the peritoneal cavity [78]. Indeed, NF-κB-mediated transcriptional activation of oncogenes, such as COX-2, inhibits apoptosis, promoting cellular proliferation in endometriosis [29,79]. When the NF-κB signalling pathway was downregulated, an alleviation of endometriosis symptoms was observed [80].

Additionally, oxidative stress triggers other pathways, including the upregulation of glycodelin, which enhances VEGF (vascular endothelial growth factor) expression and promotes angiogenesis [81].

Moreover, the activation of protein kinase ERK1/2 by ROS brings about alterations in cell proliferation and survival of endometrial cells, reminiscent of those observed in tumour cells [82]. ROS lead to oxidative protein modification, and it has been reported that women with EM show significantly higher levels of oxidative stress-associated protein in their peritoneal fluid [83]. Endometriosis-related pain has been associated with these oxidized proteins and the subsequent activation of nociceptors. Additionally, oxidized lipoproteins abundant in the endometriotic peritoneal fluid can induce MCP-1 [84]. In a recent study, a significant correlation between lipid peroxidation, ROS production, and CPP intensity was demonstrated [85]. On the other hand, the authors observed a negative correlation between the oxygen radical absorbance capacity (ORAC), expression of antioxidant capacity, and pain intensity. CPP patients exhibited elevated lipid peroxidation markers and disrupted antioxidant levels [86,87].

In endometriosis, higher numbers of macrophages are recruited to the peritoneal cavity [88]; they produce proinflammatory cytokines and prostaglandins which trigger the upregulation of nociceptive transient receptor potential (TRP) cation channels [89]. TRP cation channels have previously been linked to CPP development, given their expression in the human endometrium [90]. Oxidative stress induces changes in the transient receptor potential cation channel, subfamily V, member 1 (TRPV1), thereby contributing to pain generation in inflammatory conditions [91]. Furthermore, the activation of TRPV1 also results in increased production of ROS and an elevation in receptors for TNF-α, a recognized inducer of inflammatory hyperalgesia [92].

However, the underlying mechanisms linking endometriotic pain with oxidative stress are still unknown and require further elucidation [93].

### 3.3. Neuroangiogenesis and Neurogenic inflammation

Recent papers documented that once endometrial fragments adhere to a peritoneal region and form lesions, they promote a neuroangiogenesis process with a coordinated growth of nerves and blood vessels [94]. In detail, neuroangiogenesis is regulated by oestrogen, immune cells (especially macrophages), which are a major source of VEGF, and endothelial cells, which produce NGFs (nerve growth factors), both enhanced in EM [95]. Moreover, high levels of NGFs, as well as other neurotrophins like brain-derived neurotrophic factor (BDNF) or neurotrophin 4 and 5 (NT4/5), are found in the peritoneal fluid of women with EM, suggesting their role in the modulation of EM-associated innervation and CPP [96,97,98].

Furthermore, macrophages worsen local inflammation and are directly implicated in angiogenesis, releasing chemokines and cytokines promoting endometrial tissue growth [99,100]. While angiogenic factors (e.g., VEGF and TNF-α) enable lesion growth and formation and increase blood supply, neurotrophic factors are crucial for the proliferation of autonomic neurons and sensory afferent neurons, the latter of which can transmit nociceptive impulses [101,102]. Endometrium and endometrial lesions in EM patients have a higher density of small, unmyelinated nerve fibres (sensory afferents, sympathetic, and parasympathetic efferents) [103,104]. The great majority of these unmyelinated nerve fibres have been identified as C-fibre sensory afferents, generally operating as nociceptors that are strongly implicated in CPP development [105]. Numerous studies have shown that aberrant cytokines’ production and imbalances in sympathetic, parasympathetic, and sensory innervation can influence neurogenesis and consequent peripheral neuroinflammation in endometriosis [67]. Furthermore, women with greater nerve fibre innervation in their endometriotic lesions experienced the worst menstrual pain associated with endometriosis [3,106].

Additional studies showed that neuroangiogenesis contributes to invading and irritating the existing nerves and to deeply infiltrating endometriotic nodules [103,105,107]. Notably, endometrial adhesions and lesions can enclose or compress pelvic nerves, which contributes to CPP brought on by endometriosis [108]. Indeed, it has been demonstrated that progestogens and oral contraceptives (common hormone treatments for EM pain relief) greatly reduced the density of nerve fibres in the ectopic endometrium [104].

Finally, the phenomenon of neurogenic inflammation occurs due to the accumulation of by-products of degraded tissue, including ROS, PGE2, and acidification, which can sensitise sensory nerve fibres via receptors on nociceptive afferent nerves located within endometrial lesions in the peritoneal cavity [109,110]. The neurogenic inflammation contributes to its maintenance thanks to the release of further proinflammatory mediators, such as substance P (SP) and calcitonin gene-related peptide (CGRP) [111]. Additionally, when sensory afferent nerves are activated, mast cells are drawn to the releasing proinflammatory cytokines, such as IL-1β, TNF-α, NGF, and PGE2, leading to a persistent state of neurogenic inflammation [112]. Indeed, women with EM have higher amounts of circulating mast cells and macrophages that are located near nerve fibres and stimulated further by these mediators of inflammation [8,113].

### 3.4. Pain Sensitization

“Pain sensitization” is defined as an “increased responsiveness of nociceptive neurons to their normal input and/or recruitment of a response to normally subthreshold inputs’’ by the International Association for the Study of Pain (IASP) [114]. The sensitization manifests as allodynia (pain in response to ordinarily non-noxious stimuli) and/or hyperalgesia (abnormally heightened sensitivity to noxious stimuli) [54].

### 3.5. Peripheral Sensitization

Peripheral sensitization is defined by the IASP as an “increased responsiveness and reduced threshold of nociceptive neurons in the periphery to the stimulation of their receptive fields” [114]. Endometriotic lesions, the immune system, and peripheral nerve fibres in both the lesions and adjacent peritoneum, as well as peripheral neurons, all have a role in peripheral processes of EM-associated pain [66]. Chronic inflammation mediated by cytokines/chemokines (such as IL-1β, IL-6, TNF-α, and CCL2), growth factors (e.g., β-nerve growth factor and VEGF), and several others can increase nociceptors’ sensitivity to pain at endometrial lesions, which can cause allodynia and hyperalgesia [115]. The inflammatory environment induced in the peritoneal fluid contributes to creating a nociceptive hypersensitivity: elevated levels of TNF-α and glycodelin are correlated with a higher level of menstrual pain, altered pain response to nociceptive withdrawal reflex, and brain hyperexcitability in response to repeated electrical stimulation [112,116].

Finally, the inflammatory mediators can have the ability to directly elicit excitatory inward currents or change how ion channels work, including the TRPV1 [3]. The latter operates as a molecular sensor to enhance and integrate reactions to stimuli that cause pain, such as acidosis, oxidative stress, or inflammatory mediators [117,118]. TRPV1’s alterations have been found in other chronic pain diseases, including rheumatoid arthritis (RA), osteoarthritis, and IBS [118,119,120]. As previously reported, higher levels of ROS and neurotrophins, as found in endometriosis, promote sensitization and upregulation of TRPV1, mediating the sensitization of peripheral nociceptors, which also drives CNS sensitization [71,118,121,122].

### 3.6. Cross-Organ Sensitization

Cross-organ sensitization refers to the phenomenon where the pain in one visceral organ heightens the sensitivity to pain in another organ [54]. This interplay, particularly between the gastrointestinal, urinary, and gynaecological viscera, exemplifies another mechanism through which the peripheral nervous system contributes to mitigating EM-related pain [123]. Endometriosis is frequently associated with other pathological conditions that have been studied for cross-organ sensitization, such as IBS, inflammatory bowel disease (IBD), interstitial cystitis, and other CPP disorders [70,124].

Although the right mechanisms behind cross-organ sensitization are still unknown, it is critical to consider the overlap of peripheral afferent pathways in the dorsal root ganglion (DRG) and spinal cord. [70,125,126]. In fact, due to their physical proximity, visceral afferents converge into similar regions of the spinal cord, giving nearby cells the chance to become sensitized. Indeed, cross-organ sensitization is hypothesized to happen when the afferents that innervate one organ become more sensitive [70,127].

Due to neuroangiogenesis, the endometriotic lesions’ sensory nerves, from the peripheric afferents they initially originated from, may intersect into the same spinal pathways [128]. As a result, they will have similar central terminals inside the spinal cord and analogous cell bodies in the DRG [129]. The distribution of ectopic lesions appears to be random, which may explain why different EM patients have different pain thresholds.

### 3.7. Central Sensitization

Central sensitization is described as an increase in the nociceptive neurons’ CNS reactivity to their regular or subthreshold afferent input [114]. Although pain is commonly localized to a specific site, coordinated CNS activity leads to the conscious perception of pain originating from the brain. This phenomenon underscores the intricate nature of pain processing and perception in the CNS [130].

Neuroimaging approaches revealed that the brains of women experiencing CPP, both with and without endometriosis, exhibit alterations in regions responsible for pain perception. Specifically, a decrease in grey matter has been identified in the thalamus and insula of EM patients with CPP, but not in those who are asymptomatic. Additionally, compared to healthy women, EM patients exhibited higher amounts of excitatory neurotransmitters in their anterior insula, according to results from proton magnetic resonance spectroscopy [131].

The ongoing nociceptive process from inflamed endometriotic lesions causes long-lasting cerebral sensitization of sensory afferents. Peripheral sensitization frequently serves to both trigger and maintain central sensitization, which causes pain to persist long after the peripheral insult or illness has subsided [132]. Many women with EM report ongoing pain, despite receiving therapy for or having endometrial lesions removed. This CPP persistence, as well as the mismatch between the severity of a lesion’s diagnosis and the experienced pain, may find an explanation in the central sensitization of pain pathways. This phenomenon suggests that changes in the CNS processing of pain signals contribute to the ongoing perception of pain, independent of the presence or severity of physical lesions [133,134].

The relationship between endometriosis and changes in central pain processing raises the question of whether these changes exacerbate CPP due to alterations caused by EM or if women with these changes are inherently more sensitive to endometrial disease. The precise causal direction of this connection remains unclear. However, the prolonged period between the pain onset and the EM diagnosis provides ample time for these lesions to induce the chronic alterations necessary for triggering central sensitization. In addition, animal models documented that removing lesions in the early stages led to reduced pain experience [135,136].

Additionally, CPP is frequently linked to detrimental cognitive, behavioural, sexual, and emotional outcomes, potentially aggravating pain [33]. The interaction between psychological discomfort and pain perception is bidirectional, as psychological factors can influence the perception of pain, while pain itself may contribute to psychological discomfort [56].

## 4. Impairment of the Immune System in Endometriosis

As previously mentioned, several studies highlight an immune system dysfunction in EM patients. Indeed, the relative and absolute levels of various immune cells and factors that play key regulatory roles are altered in the peritoneal fluid and the endometriotic lesions [2,137,138]. In general, EM etiopathology involves the participation of both innate (e.g., macrophages, neutrophils, mast cells, dendritic cells (DCs), and natural killer (NK) cells) and adaptive immune cells (T and B cells) [139,140].

Concerning innate immune cells, macrophages and neutrophils are initially recruited to the peritoneal cavity in response to inflammation caused by menstrual endometrial fragments. Specifically, these cells secrete and promote the release of proinflammatory cytokines and angiogenic mediators, like TNF-α, IL-17, and IL-8 [141], while also modulating hypoxia-induced angiogenesis through the VEGF secretion [142]. The presence of these cytokines may induce a prolonged state of chronic inflammation. Macrophages can directly induce excitatory alterations in TRPV1 channel activity, encourage the sensitization of peripheral nerves, provoke sensitivity, and start a complex feedback loop that enhances microenvironmental inflammatory reactions and exacerbates pain generation. Elevated levels of TRPV1/TRPA1 isomers may prompt macrophage polarization, facilitate the migration of ectopic endometrial cells [143], and subsequently advance EM development. Increased IL-17 levels within the peritoneal fluid regulate the recruitment and polarization of macrophages toward the alternatively activated M2 phenotype (anti-inflammatory) as opposed to the M1 phenotype (proinflammatory). Additionally, they promote angiogenesis, thereby stimulating the growth of endometriotic lesions [144]. Therefore, alterations in the inflammatory profile within the abdominal microenvironment prompt macrophages to secrete elevated levels of cytokines. This, in turn, leads to the recruitment of greater numbers of macrophages and other immune cells. Consequently, this condition promotes the migration and invasion of endometrial cells favouring the development of endometriotic lesions. Furthermore, neutrophils produce IL-17α, the level of which is higher in EM patients and connected to the severity of the disease and infertility [145,146,147], and release VEGF in the abdominal cavity, promoting the development of endometrial lesions [148,149]. Activated mast cells directly contribute to the symptoms of neuropathic pain by producing mediators such as histamine, leukotrienes, tryptase, TNFα, PGs, serotonin, IL-1, and IL-8. The recruitment of leukocytes that produce analgesic mediators by the activation of mast cells may also indirectly contribute to the rise of neuropathic pain [150].

The endometriotic lesions and the surrounding peritoneal membrane show the presence of immature DCs, while mature DCs are diminished throughout the menstrual phase [151]. In addition, NK cells are diminished within the peritoneal cavity. NK cells have significant roles in EM development, either promoting or contrasting the survival, implantation, and proliferation of endometrial cells [152]. NK cells are also implicated in various pathogenic processes involving immune cells by modulating cytokine expression levels [153]. Moreover, NK cells may contribute to the onset of clinical symptoms like dysmenorrhea and pelvic discomfort, as well as complications (e.g., infertility) in EM patients [154].

Regarding adaptive immunity, its role in EM is complex and contrasting, as it is influenced by the phases of the disease. In detail, T cells are increasingly recognized as significant factors in EM pathogenesis. Patients with endometriosis showed T helper (Th) cells towards the Th2. This is attributed to intracellular expression of IL-4 and the absence of IL-2 in lymphocytes isolated from ectopic lesions [155]. Additionally, T cell function may be correlated with EM severity [156]. A notable accumulation of Tregs (regulatory T cells) in the peritoneal fluid of patients with advanced endometriosis has been observed [157]. This accumulation may be associated with persistent local inflammatory responses and the chemotaxis of inflammatory cells. Insufficient local induction and activity of Tregs in the early EM stages might adversely affect the function of other effector immune cells (such as macrophages, neutrophils, and NK cells), thus collectively promoting EM persistence and progression [157]. Reduced Tregs may decrease the aggregation of platelets, M2 macrophages, and Th2 and Th17 cells and increase that of Th1 cells in lesions, thereby inhibiting epithelial–mesenchymal transition (EMT), fibroblast to myofibroblast transdifferentiation (FMT), and fibrosis, consequently impeding EM progression [138].

Moreover, heightened activation of B cells has been detected in both the eutopic endometrium and the lesions compared to the healthy endometrium [158].

Finally, endometriotic lesions can also increase the expression of PGs, MCP1, glycodelin, and other inflammatory mediators and pain-related compounds [159,160]. These inflammatory and pain-associated factors influence inflammatory cells in turn. Consequently, the recruitment of more inflammatory cells induces a vicious cycle, leading to the establishment of a new impaired inflammatory milieu in the peritoneal and pelvic environment [139].

Regarding the molecular immune response, the peritoneal fluid harbours elevated concentrations of proinflammatory and angiogenic cytokines, originating from macrophages as well as from the lesions. Several studies have highlighted increased concentrations of IL-1α [161], IL-1β [162], and total IL-1 [163], supporting the presence of a localized inflammatory milieu. Moreover, endometriotic lesions exhibit higher levels of IL-1β expression compared to both eutopic endometria from healthy women and those with endometriosis, underscoring the locally induced inflammation characteristic of endometriosis. The overproduction of TNF-α, induced by activated macrophages, NK cells, and Th1 cells, has been observed in the peritoneal fluid exclusively in the mild or early stages of the disease [163], suggesting its involvement during the initial EM phases. Both TNF-α and IL-1β stimulate the expression of cyclooxygenase-2 (COX-2), responsible for regulating the synthesis of prostaglandin E2 (PGE2) [164]. Additionally, PGE2 can induce COX-2 expression, establishing a positive feedback loop that amplifies inflammation and pain through excessive PGE2 production. In addition, PGE2 can also diminish macrophage cytotoxicity and stimulate cell proliferation, local oestrogen synthesis, and angiogenesis [164]. Besides, elevated levels of IL-6 produced by macrophages, Th1 cells, B cells, fibroblasts, and endothelial cells were found in the peritoneal fluid of EM patients [165], positively correlated with the size and number of endometriotic lesions [165]. Furthermore, IL-6 concentrations increase in more advanced EM stages [166]. The increase in IL-1β and TNF-α could stimulate peritoneal mesothelial cells to produce IL-6, thereby exacerbating the local inflammation observed in endometriosis. Moreover, high IL-10 levels have been observed in the peritoneal fluid of EM patients [167] compared with healthy subjects, as well as women with other gynaecological diseases [168]. The increased concentration of IL-10 has been linked to the reduced cytotoxicity of NK cells observed in endometriosis [169], underscoring the concept that local cytokine dysregulation facilitates the implantation of endometrial fragments in the peritoneal cavity. Additionally, elevated levels of the neutrophil chemotactic IL-8 [88] have been detected in the peritoneal fluid of EM women [170], although not in the serum or peripheral blood [171], suggesting a localized dysregulation in endometriosis. Moreover, higher IL-8 levels have been reported in the early EM stages compared to more advanced stages [171]. Finally, MCP-1 has been detected in high concentrations in the peritoneal fluid of EM women [170], increasing with disease severity [172]. Peritoneal mesothelial cells of women with endometriosis produce MCP-1 in response to IL-1α and TNF-α stimulation. In healthy women, MCP-1 production was correlated with the stage of the menstrual cycle, where the peritoneal fluid of healthy women had higher MCP-1 levels during the proliferative phase compared to the secretory phase. These results point towards a responsiveness of MCP-1 to ovarian hormones. The production and expression of MCP-1 in isolated endometrial stromal cells are inhibited by E2 in a dose-dependent manner [173]. Endometriotic lesions can be stimulated to produce MCP-1 by IL-1β, and this response is further enhanced by E2. These results not only demonstrate the significant involvement of MCP-1 in EM development but also highlight the complex interplay between the endocrine and immune systems, showing the crucial role of oestrogen in amplifying the chemokine-induced recruitment of immune mediators to the sites of endometriotic lesions.

## 5. Microbiome Composition and Function in Endometriosis

Currently, researchers are investigating the potential role of the microbiome as a counterpart of immune response in EM development. This is because certain bacteria can stimulate the immune system and induce inflammation, while others contribute to the host’s homeostasis by producing antimicrobial or immunomodulatory compounds. The microbiome may indeed contribute to EM development by promoting hormonal dysregulation (through the estrobolome, defined as the microbiome members capable of metabolizing and modulating oestrogen), altering cellular proliferation/apoptosis, metabolism, oxidative stress, and angiogenesis [174]. Moreover, recent investigations demonstrated the potential interaction between the microbiomes of various human disorders, such as the gut and urogenital microbiomes, given their proximity [175,176].

### 5.1. The Gut, Vaginal, and Peritoneal Bacterial Flora

Differences in the faecal microbiome composition between women with endometriosis and those without showed the depletion of several taxa, including *Lachnospiraceae Ruminococcus, Clostridia Clostridiales*, *Ruminococcaceae Ruminococcus*, and *Clostridiales Lachnospiraceae*, along with an increased abundance of *Eubacterium dolicum* and *Eggerthella lenta* in EM patients as compared to EM-free women [177]. Likewise, an increased *Firmicutes/Bacteriodetes* ratio in the endometriosis group was reported, with enrichment of *Cynaobacteria*, *Actinobacteria*, *Fusobacteria*, *Saccharibacteria*, and *Acidobacteria* as compared to the control [177]. In addition, the LEfSe analysis demonstrated that *Blautia*, *Dorea*, *Bifdobacterium*, and *Streptococcus* abundances are related to the inflammatory and serum hormone levels [177]. Another study found that *Firmicutes* and *Bacteroidetes* were the major abundant phyla in the gut microbiomes of EM patients. Furthermore, they found that *Bacteroidetes*, *Proteobacteria*, *Actinobacteria*, *Firmicutes*, *Fusobacteria*, and *Verrucomicrobia* in the gut were correlated with concentrations of urinary oestrogen [178]. In another research, both the alpha and beta diversities were different between EM patients and healthy controls [179]. In addition, *Coriobacteriia, Bacilli*, *Clostridia*, *Bacteroidia*, and *Gammaproteobacter* levels differed between the endometriosis group and the control group. A significant variation in β diversity was additionally observed in experimental animals. *Bacteroides* were enriched in the control group while *Firmicutes* were enriched in the model group [179]. The *Firmicutes/Bacteroides* ratio in the body also increased in rats with EM, suggesting that endometriosis causes a gut microbiome imbalance [180].

Furthermore, a decrease in the synthesis of microbial products such as short-chain fatty acids (SCFAs) has been directly connected to an imbalance in the gut microbiome [181,182]. In detail, SCFAs, like butyric acid, are crucial for preserving the intestinal barrier, reducing immunological response, and enhancing mitochondrial performance [183]. In addition, butyrate acts to avoid the immune response caused by the gut biological imbalance [184]. Finally, a recent study reported that the gut microbiome can be linked to depression via reducing the butyrate; indeed, as previously mentioned, depression is more common among EM patients, and women who experience persistent pelvic discomfort typically experience depression at a higher level than those who do not [185]. A recent study demonstrated that butyrate produced from the gut microbiome protects mice from endometriosis via controlling G-protein-coupled receptors [186].

In mice with endometriosis, a study examining the relationship between faecal metabolomics and gut microbiome showed a decrease in linolenic acid (ALA) abundance and a rise in chenodeoxycholic acid (CDCA) and ursodeoxyl content [187]. Another report demonstrated that ALA reduced the inflammatory response by inhibiting the accumulation of nitrite and prostaglandin E2 (PGE2) [188]. Additionally, in EM mice, ALA can lessen the LPS amount and enhance the inflammatory milieu in the abdomen [188]. Collectively, all these findings essentially suggest that the gut microbiome plays a role in the EM onset; in detail, metabolic changes can participate in the disease pathogenesis, and some microbial metabolites may have the potential to treat endometriosis.

On the other hand, an increasing body of research indicates that EM women have higher levels of bacterial colonization in their menstrual blood and endometrial tissues than do women in the general population [189,190,191,192].

A different distribution of microorganisms in the various sections of the female reproductive tract has been reported [193,194]. Notably, *Lactobacillus* spp dominates the lower tract, creating a low-pH environment and producing bacteriocins as well as hydrogen peroxide, thus protecting the host against pathogens. Studies of the vaginal microbiome using the community state type (CST) classification system reported five CSTs that can change throughout women’s lifetimes, whereby CST I, II, III, and V are dominated by *Lactobacillus crispatus*, *Lactobacillus iners*, *Lactobacillus gasseri*, and *Lactobacillus jensenii*, respectively [195,196]. These four CSTs are associated with healthy vaginal microbiome composition, whereas CST IV, which exhibits greater percentages of strictly anaerobic microorganisms (e.g., *Finegoldia*, *Prevotella*, *Atopobium*, *Dialister, Aerococcus*, *Gardnerella*, *Peptoniphilus*, *Megasphaera*, *Sneathia*, *Eggerthella*, and *Mobiluncus*), is hypothesized to be connected to vaginal inflammation or dysbiosis. However, despite *Lactobacillus* spp. dominating the microbiome ecosystem, according to a recent study, EM patients also have a larger abundance of *Corynebacterium*, *Enterobactericaea*, *Flavobacterium*, *Pseudomonas*, and *Streptococcus* in their cervical microbiomes than controls [197].

Additional research has demonstrated that a high concentration of *Gardnerella*, *Prevotella*, and *Bacteroides* spp. in the cervicovaginal microbiome may increase the risk of endometriosis and other pelvic inflammatory diseases. Consequently, this could lead to infertility [198].

Moreover, the American Society of Reproductive Medicine (ASRM) classification system divides endometriosis stages into four grades according to the number of lesions and depth of infiltration: minimal (Stage I), mild (Stage II), moderate (Stage III), and severe (Stage IV). Several differences in the microbial vaginal architecture have been observed between EM patients in Stages I and II as compared to those in Stages III and IV [199], and potential vaginal microbial biomarkers can be specific for different stages: (a) Stage I–II: *L. jensenii* or members in *Corynbacteriales*, *Porphyromonadaceae*, and *Ruminococcaceae*, (b) Stage III–IV: *Bifidobacterium* breve and *Streptococcaceae* members. In addition, the metagenome activities analysed through bioinformatic tools showed a higher proportion of bacteria involved in general and lipid metabolism, as well as the synthesis and breakdown of ketone bodies [200].

Finally, the peritoneal microbiome has also been explored to elucidate EM development. This tract was once thought to be “sterile”; however, a recent study reported a total of 276 operational taxonomic units (OTUs) detected in peritoneal fluid collected from EM patients (as compared to 211 OTUs in the control group), out of which 120 were typical of the endometriosis group [201]. At the genus level, there was a significantly higher abundance of *Acidovorax*, *Devosia*, *Methylobacterium*, *Phascolarctobacterium*, and *Streptococcoccus* in the EM group than in the control group. Moreover, microbes present in the extracellular vesicles in the peritoneal fluid of EM patients showed a significant decrease in *Actinobacteria* [202]. However, a recent investigation demonstrated that the prognostic value of the vaginal microbiome in endometriosis might not be as relevant as that of the gut microbiome, which to some extent brings a new direction in EM exploration [177]. In peritoneal and intestinal fluids, *Ruminococcus* and *Pseudomonas* have been suggested as potential biomarkers for EM diagnosis [177].

### 5.2. The Oestrogen–Gut Microbiome Axis and the Estrobolome

As previously mentioned, endometriosis is an oestrogen-dependent disease. In detail, by raising mucus secretion, glycogen levels, and epithelial thickness, oestrogen can control the microenvironment of the female lower genital tract. It can also indirectly lower vaginal pH by raising lactic acid levels and *Lactobacillus* abundance [203]. The host metabolism of oestrogen mainly occurs in the liver. Indeed, the liver can produce sex hormone-binding globulin, and the combination of sex hormone-binding globulin and oestrogen can lead to the loss of oestrogen biological activity [204]. Plottel and Blaser defined “estrobolome” as “the aggregate of enteric bacterial genes whose products are capable of metabolizing estrogens” [205]. Indeed, the gut microbiome produces β-glucuronidase and β-glucosidase, and these products can promote the degradation of oestrogen, thus increasing the reabsorption of free oestrogen and improving the level of oestrogen in circulation [206,207]. Distinct bacterial ß-glucuronidase genes from the human gut microbiome have been reported [208,209]. The well-characterized *gus* gene is commonly found in gut bacteria here, as the *BG* gene has been reported by metagenomic analysis [209]. The *BG* gene is represented in the bacterial phyla *Bacteroidetes* and *Firmicutes*. On the other side, *gus* is more common in *Firmicutes* [210].

Multiple bacterial genera in the gut microbiome can produce β-glucuronidase, including *Bacteroides*, *Bifidobacterium*, *Escherichia coli*, and *Lactobacillus* [211]. It has been reported that EM patients have a faecal increase in *Escherichia coli* [190,192]. Collectively, all these studies confirmed that the gut microbiome could lead to an increase in circulating oestrogen levels, which could induce a high-oestrogen environment for EM progression [212]. Of course, further research is needed to discover the factors that stimulate the production of β-glucuronidase by specific gut microbiomes in EM pathogenesis, as well as the relationship between the gut microbiome and the female upper reproductive tract microbiome and whether they collaboratively operate to cause the disease.

## 6. Microbiome, Inflammation, and Oxidative Stress as Modulators of CPP Associated with Endometriosis

As previously reported, endometriosis is characterised by changes to the microbiome architecture in the gut, female reproductive tract, and peritoneal fluid. It is not yet clear whether these changes are the cause of EM or consequences [213]. However, the microbiome plays a central role in the modulation of the inflammatory, immune, and oxidative processes of endometriosis [28] (Figure 2).

In healthy conditions, the gut microbiome contributes to the regulation and maintenance of the mucus thickness, transepithelial barrier function, and immunological response of the epithelium [174]. On the other hand, dysbiotic gut bacteria frequently cause an increase in both local and systemic inflammation by affecting the intestinal protective mucus layer and directly interacting with enterocytes [214]. Additionally, dysbiosis can cause intestinal barrier disruption and immunological dysfunction, which can result in the transmigration of the gut bacteria [174]. Moreover, microbial dysbiosis and infections in the female genital tract have the potential to trigger genetic and epigenetic events. These conditions may contribute to elevated oxidative stress, characterized by an increase in ROS and alterations in immune responses, which may influence EM development [215]. Therefore, gut dysbiosis, which occurs in endometriosis, contributes to higher levels of systemic inflammation but also to infertility [216].

In addition, EM-associated CPP and the gut microbiome seem to be related. A recent study showed that patients with CPP exhibited less diversity in their gut microbiome than the control group [217]. Moreover, the female reproductive tract’s microbiome also appears to be involved. Notably, the cervicovaginal microbiome’s ascent to the upper female reproductive system and gut–vagina axis (GVA) may contribute to the dysbiosis spread from the gut to the cervicovaginal microbiome, as well as inflammation, which could contribute to endometriosis, infertility, CPP, and symptoms related to these illnesses [218,219].

Direct and indirect processes allow the gastrointestinal tract to communicate with the central nervous system in both directions. This complex interaction is referred to as the gut–brain axis (GBA) [220]. It has been reported that gut bacterial products can change the host’s brain activity when they contact with host receptors through the action of neuroactive compounds, such as serotonin, GABA, glutamate, and SCFA [220,221,222]. This interaction can affect neuronal transmission, pain creation, inflammation, or hormone release [223].

Regarding the relationship between microbiome and pain, microorganisms that influence brain activities could increase nociceptive transmission, thus mediating CPP. In detail, the microbiome influences the synthesis of compounds that affect multiple neurons both centrally and peripherally, moderating the nociceptive pain. Furthermore, neuroimmune activation that may be controlled by the microbiome appears to be the cause of neuropathic pain, which frequently results from nerve-damaging trauma and gradually leads to central sensitization of chronic pain [219]. Finally, the gut microbiome affects microglial function, but it also regulates other cells, such as monocytes, astrocytes, endothelial cells, microglia, macrophages, pericytes, and T cells, leading to the release of proinflammatory chemokines/cytokines (CCL2, CXCL-1IL-1β, IFN-γ, and TNFα) [223]. These elements act on synaptic neurotransmission by increasing glutamate levels and lowering GABA (gamma-aminobutyric acid) levels, but they also affect serotonin and SCFA levels, thus leading to central sensitization and hyperalgesia [224,225].

Furthermore, these metabolites move into the brain and bind to neural receptors. This bond directly activates cerebral and hypothalamus neurons, including GnRH neurons, modulating GnRH secretion from the hypothalamus [226]. Since GnRH neurons have been observed to innervate outside of the blood–brain barrier (BBB), it is possible that neuroactive substances can interact with GnRH neurons without passing through the BBB [227]. GnRH acts on the pituitary gland, causing the release of luteinising hormone (LH) and follicle-stimulating hormone (FSH) into circulation, which stimulates oestrogen secretion and follicular development in the ovaries [228]. Gut dysbiosis may dysregulate the hypothalamus–pituitary–ovarian axis (HPO) and consequently, LH and FSH production, contributing to central and visceral pain generation [219,226]. Consequently, LH dysregulation can affect oestrogen production, leading to cervicovaginal dysbiosis and increasing the risk for oestrogen-related disorders, including endometriosis, infertility, and CPP.

Finally, gut dysbiosis can also dysregulate the hypothalamus–pituitary–adrenal axis (HPA), increasing the secretion of corticotropin-releasing hormone (CRH) and consequently, cortisol. Therefore, higher cortisol levels can contribute to systemic inflammation and lower pain tolerance, leading to CPP [219]. So, dysregulation of HPA axis signalling may contribute to mood disorder, depression, and anxiety, which are frequently associated with elevated cortisol levels and inflammatory mediators that cause a protracted inflammatory state [229].

## 7. Microbiome-Mediated Inflammation and Its Link with Endometriosis-Related Psychological Disorders

The link between the microbiome and psychological disorders is becoming clearer [230]. As previously stated, the gut microbiome can influence brain activity through serotonin, glutamate, GABA, and SCFA, as well as by modulating noradrenergic and dopaminergic neurotransmission, thereby impacting neuropsychological aspects [231]. However, numerous studies have pointed to inflammation, a counterpart of the microbiome, as a crucial element in the aetiology of anxiety and depressive disorders. Moreover, inflammation and oxidative stress are connected in complex ways to impaired neuroplasticity and thus to the development of bipolar [232] and psychiatric disorders, including depression [233] and psychosis [234]. At least some of these proinflammatory changes have also been observed in EM patients [2].

In fact, several studies have demonstrated that patients with anxiety or depression, despite being otherwise healthy, exhibit higher levels of acute phase proteins and proinflammatory cytokines [235,236,237]. Similarly, the immunity dysregulation observed in mental health problems may be influenced by alterations in the gut microbiome and the subsequent cascade of proinflammatory and oxidative responses.

Through signals across the BBB and immune cells leaking into the brain, peripheral inflammation may be linked to brain function [235]. Additionally, bacterial metabolites and lipopolysaccharide infiltration may activate innate resistance receptors, leading to CNS inflammation. It is believed that longer-term activation of this system interferes with the HPA axis’ ability to downregulate hormone levels, which results in persistently high cortisol levels in the blood and, as such, in anxiety and depression [238].

The contribution of microbiome-mediated inflammation to anxiety and depression is suggested by the increase in inflammation-associated microbial members and the decline in species that generate anti-inflammatory metabolic products [239], such as species that secrete anti-inflammatory SCFAs, including *Faecalibacterium*, *Coprococcus*, and *Clostridium XIVa* species that can produce butyrate [238] and *Megamonas* producing acetate and propionate [240]. Reduced SCFA-producing species and its products may be a factor in immune system dysregulation [239]. Indeed, as previously mentioned, lower butyrate levels have also been linked to greater intestinal permeability [241]. As a result, microbial metabolites and endotoxins such as lipopolysaccharides can translocate into the bloodstream, affecting organs throughout the body [242]. Moreover, gram-negative bacteria can enter the bloodstream when intestinal permeability is increased by high amounts of circulating cortisol and inflammatory mediators, which may lead to chronic CNS inflammation and contribute to CPP [243,244]. Therefore, this evidence suggests that the microbiome may play a role in EM, CPP development, and related psychological comorbidities.

## 8. Regulating the Microbiome, Inflammation, and Oxidative Stress through Diet: A Strategy for Managing CPP and EM-Associated Psychological Disorders 

The hypothesis that appropriate nutritional interventions can influence EM progression is based on observations that diet and nutritional supplements can impact various molecular processes, such as oestrogen activity, prostaglandin metabolism, and inflammation [1], as well as microbiome composition and function [245]. For example, both diet- and nutrition-related changes in the gut microbiome architecture influence inflammation and the GBA, which, in turn, may then affect endometriosis-associated symptoms, including CPP. Consequently, this effect may act on psychological manifestations such as anxiety and depression, either indirectly through regulating the underlying processes of CPP or directly through the modulation of the GBA and HPA axis [239].

### 8.1. Dietary Intervention in Endometriosis

Currently, diverse dietary strategies are suggested for mitigating EM-associated symptoms and pain, though only a limited number of low-quality studies have explored their effectiveness. Overall, regardless of the specific type of dietary intervention advised, an enhancement in symptomatology tends to be observed. In a small Swedish study comprising twelve individuals with endometriosis, participants reported enhanced well-being and reduced symptoms following dietary and lifestyle changes, underscoring the substantial diet impact on their health [246]. The study highlights the efficacy of personalized dietary interventions in addressing EM symptoms, emphasizing the crucial role of involving patients in treatment decisions for optimal outcomes [246].

In addition, diets high in soluble fibre that boost a microbiome rich in butyrate-producing species have been linked to decreased levels of both proinflammatory cytokines and anxiety manifestations [247]. Women with EM responded to the low fermentable oligosaccharides, disaccharides, monosaccharides, and polyols (FODMAP) diet, which is used as therapy for IBS patients [248]. In detail, FODMAPs are short-chain carbohydrates (mainly found in grains, fruits, and vegetables) that are easily fermented by the microbiome and are not properly absorbed in the small intestine. These carbohydrates have an osmotic effect and determine gas production that contributes to abdominal distention. Because of this, patients with visceral hypersensitivity (present in both IBS and endometriosis patients) experience pain and bloating. Women with endometriosis and IBS who avoided FODMAPs reported less pain. However, it remains unclear whether this pain relief specifically targeted endometriosis-related symptoms [248]. The low-FODMAP diet, comprising three stages—restriction, reintroduction, and personalization—has been suggested as a safe long-term approach under the guidance and support of a clinical dietitian [249]. Moreover, in pre-clinical investigations, the low-FODMAP diet changed the gut microbiome, which decreased the amount of gram-negative bacteria lipopolysaccharide (LPS) in the faeces [250]. Furthermore, *Akkermansia muciniphila* and *Actinobacteria* exhibited significant decreases compared to a high-FODMAP diet. Consequently, adopting a low-FODMAP diet may contribute to decreasing gut mucosal inflammation, promoting the repair of barrier function, and ultimately alleviating visceral pain [250].

On the other side, plant-based nutrition appears to be linked to an anti-inflammatory profile. Vegetal foods, which are rich in polyphenols, have the potential to reduce inflammation when converted into bioactive chemicals. Furthermore, individuals following a plant-based diet display heightened levels of anti-inflammatory substances in their gut microbiome compared to those following an omnivorous diet [251]. In a randomized crossover trial with women experiencing dysmenorrhea, the implementation of a low-fat vegetarian diet demonstrated an increase in plasma sex-hormone-binding globulin expected to lower oestrogenic activity. Additionally, this dietary approach has proven effective in reducing pain intensity and duration and managing premenstrual signs compared to an omnivorous diet [252].

Furthermore, meat consumption has been associated with an increased EM risk [253]. Red meat may elevate proinflammatory markers and contribute to higher levels of oestradiol and estrone sulphate, potentially leading to increased steroid levels, inflammation, and EM onset [253,254].

Moreover, the Mediterranean diet (MD) offers numerous benefits in the treatment of gynaecological conditions and serves as a preventive measure against various non-communicable diseases, including cancer and cardiovascular disease [255,256]. According to a single-arm study, adopting a diet rich in fresh products, whole grains, soy, fish with fat, white meat, and extra virgin olive oil while reducing intake of red meat, sugary drinks, animal fats, and sweets resulted in improved overall well-being for EM patients. Additionally, this dietary approach was associated with a decrease in pain, dyspareunia, dysmenorrhea, and dyschezia [257]. Interestingly, extra virgin olive oil, a cornerstone of the MD that is rich in oleocanthal, shares a structural similarity with the drug ibuprofen. Both substances exhibit the ability to inhibit the enzyme cyclooxygenase. Furthermore, the enhanced fibre content of this diet contributes to a positive impact on digestion (eupeptic effect), while magnesium-rich diets may potentially delay the increase in intracellular calcium levels, thereby hastening uterine relaxation [258].

The MD is well known for its antioxidant and anti-inflammatory effects, which may contribute to a decrease in EM-associated pain [85]. Cirillo et al. demonstrated MD efficacy in alleviating pain associated with dyspareunia, non-menstrual pelvic pain, dysuria, and dyschezia. Additionally, their study highlighted a positive correlation between lipid peroxidation and the severity of non-menstrual pelvic pain and dysuria measured by the VAS, as well as a significant negative correlation between ORAC and the severity of non-menstrual pain and dyschezia [85]. Mier-Cabrera et al. compared antioxidant intake between women with and without endometriosis and evaluated the impact of a high-antioxidant diet (HAD) on oxidative stress markers. EM patients exhibited lower intake of vitamins A, C, E, zinc, and copper compared to their counterparts without the condition [259]. The HAD intervention, featuring elevated levels of vitamins A, C, and E, resulted in increased antioxidant markers and decreased oxidative stress markers in EM women. These data suggest the potential benefits of dietary interventions rich in antioxidants for managing oxidative stress in endometriosis [259].

Furthermore, a gluten-free diet may alleviate pain by inhibiting gluten-mediated immunomodulation and the inflammatory response, affecting the cytokines’ profile. In a retrospective study on a gluten-free diet, 75% of EM patients reported significant improvement in their pain problems [260]. An additional work investigated the impact of a low-nickel diet, considering the high prevalence of nickel allergic contact mucositis in EM patients. The authors documented a significant decrease in gastrointestinal, extra-intestinal, and gynaecological symptoms [261].

Finally, EM women show a higher prevalence of food restrictions, primarily due to allergies, intolerances, and, most notably, gastrointestinal symptoms, in comparison to controls [262].

Tailored nutritional interventions administered by trained dietitians have the potential to alleviate disease burden and enhance quality of life for EM patients; surely, further research is essential to establish evidence-based dietary recommendations for effective EM management.

### 8.2. Nutritional Supplementation

Various studies evaluated the effect of nutritional supplementation on microbiome, inflammation, oxidative stress, and associated pain. Regarding probiotic supplementation, there are limited studies on its impact on EM physical and psychological symptomatology. Specifically, two mouse models examined the impact of oral supplementation with *Lactobacillus gasseri*, revealing its potential to prevent EM development and progression. The mechanism proposed involved immunostimulatory action through NK cells’ activation and a decrease in the growth of ectopic endometriotic lesions [263,264]. Furthermore, the administration of *Lactobacillus gasseri* OLL2809 appeared to be effective in pain treatment. Indeed, it has been reported that the intensity of pain during the menstrual period, measured through the VAS and the verbal rating scale (VRS) of dysmenorrhea, was significantly improved in treated EM patients when compared to the placebo group [265]. Additionally, the supplementation of *Lactobacillus* sp. may have had some positive effects on EM patients’ related pain when compared to the control group [266].

Nonetheless, additional research is required to evaluate the specific role of probiotics and prebiotics in addressing dysbiosis, thereby bolstering the rationale for incorporating microbiome-based therapies in EM management.

Dietary supplementation has been suggested as a novel approach to alleviate pain-associated symptoms [267,268]. A study on EM patients evaluated a 3-month treatment with a blend of quercetin, curcumin, parthenium, nicotinamide, 5-methyltetrahydrofolate, and omega-3 and omega-6. Using a Visual Analog Scale (VAS), the authors noted a significant reduction in symptoms among the EM-treated patients compared to the control group [268]. Sesti et al. demonstrated that postoperative dietary supplementation, encompassing vitamins, mineral salts, VSL3 lactic ferments, omega-3, and omega-6, along with postoperative hormonal suppression therapy, proved more effective than surgery coupled with a placebo in relieving pain associated with stage III–IV endometriosis and improving overall quality of life [267]. Moreover, it is well accepted that polyunsaturated fatty acids (PUFAs) like omega-3 and omega-6 have a therapeutic anti-inflammatory effect on a variety of disorders, such as cardiovascular disease and metabolic syndrome [269]. Currently, the evidence of the beneficial effects of PUFAs on endometriosis is growing, and it has been reported that women with higher omega-3 PUFA intake have a lower EM risk [270,271]. Additionally, studies on a murine model have shown that mice with high amounts of endogenous omega-3 PUFAs have decreased IL-6 levels. Additionally, when donor tissue was transplanted into a host environment abundant in PUFAs, there was an observed decrease in the growth of endometriosis-like lesions.

Omega-3 PUFAs can also influence the immunological, angiogenic, and proliferative factors involved in early EM development [272,273]. Several studies reported that omega-3 could reduce dysmenorrhea symptoms [274,275]. Furthermore, women with dysmenorrhea who had high omega-3/omega-6 ratios reported less pain [275]. Finally, in three clinical trials, palmitoylethanolamine (endogenous fatty acid amide) and transpolydatin (natural glucoside of resveratrol) were administered to EM patients. These studies reported that the VAS score of the intervention group for EM-related pain was lower than the VAS score of the control group [276,277,278].

Furthermore, EM prevention and treatment may benefit from adequate vitamin D intake; indeed, low vitamin D levels were linked to a higher probability of EM diagnosis and more severe symptoms [279]. In a 12-week, randomized, placebo-controlled study, EM patients treated with vitamin D every two weeks showed a decrease in their self-reported level of pelvic pain. Additionally, vitamin D was linked to diminished levels of high-sensitivity C-reactive protein, as well as increased levels of total antioxidant capacity, suggesting that it may have both an antioxidant and anti-inflammatory effect [280]. On the other hand, Almassinokiani et al. showed that vitamin D is not effective in EM-related pain treatment [281].

Recently, it has been demonstrated that the use of antioxidant vitamin supplementation is generally effective in reducing EM-related pain and inflammatory markers [282].

The effects of diet and dietary supplementation on endometriosis-related pain, microbiome, inflammation, and oxidative stress are documented in Table 1, as reported in various studies.

## 9. Conclusions and Future Directions

To conclude, there is a complex relationship among endometriosis, pain, and psychological comorbidities such as stress, anxiety, neurological disorders, and depression. In truth, psychological symptoms usually coexist with physical ones, with CPP serving as a key mediator between endometriosis and these comorbidities. In addition, as the EM disease is characterized by chronic inflammation, it is crucial to delve into the role of the main modulators of inflammation, such as the microbiome (and its metabolites) and oxidative stress, in regulating CPP as well as psychological comorbidities. Recent evidence suggests that the microbiome influences pain modulation in both the peripheral nervous system and the CNS. Therefore, in the context of the existing literature, we propose that regulating bacterial flora via dietary adjustments and probiotics/prebiotics or similar interventions could offer a new therapeutic avenue for addressing CPP and psychological disorders. However, it is critical to note that while the research in this field is promising, it is still in its early stages. The previous studies have several limitations, including heterogeneous study designs, small sample sizes, absence of control groups, and a lack of prospective research. More comprehensive and consistent clinical trials are needed to establish the safety and especially the efficacy of such interventions. Elucidation of the interplay among the microbiome, inflammation, and oxidative stress in endometriosis will be useful for limiting pain and improving the quality of life and mental health of patients. The new findings will be relevant and encouraging for the gynaecological community in the development of clinical and nutritional intervention protocols to maintain and support the mental well-being of patients.

## Figures and Tables

**Figure 1 ijms-25-06473-f001:**
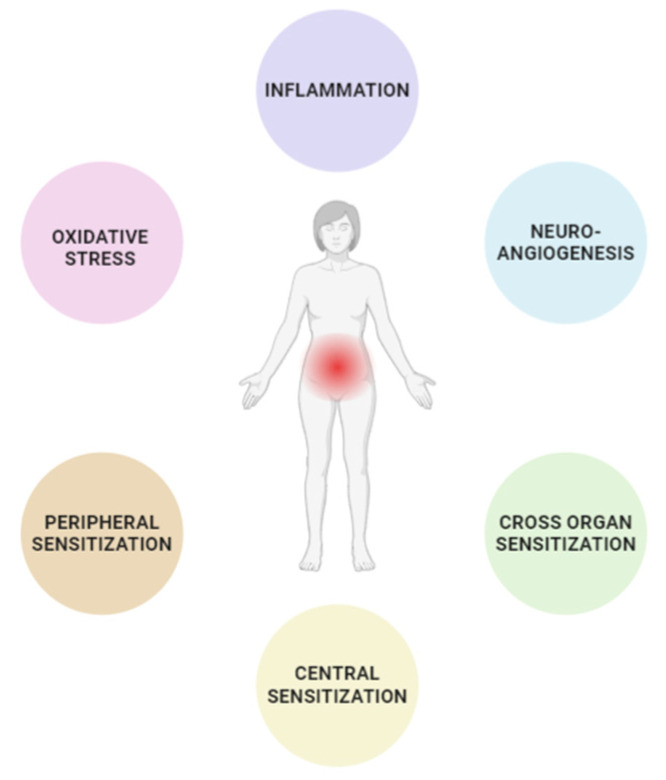
Principal mechanisms involved in Chronic Pelvic Pain (CPP) onset.

**Figure 2 ijms-25-06473-f002:**
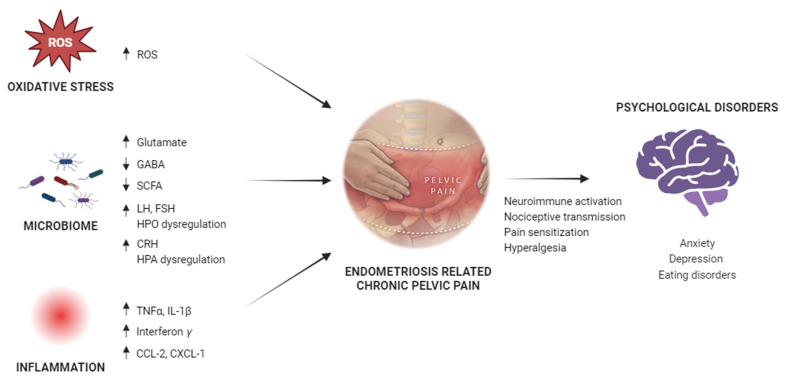
The influence of oxidative stress, microbiome, and inflammation on CPP in endometriosis and their impact on related psychological disorders. ROS: reactive oxidative species; GABA: gamma-aminobutyric acid; SCFA: short-chain fatty acid; HPO: hypothalamus–pituitary–ovarian axis; LH: luteinising hormone; FSH: follicle-stimulating hormone; CRH: corticotropin-releasing hormone; HPA: hypothalamus–pituitary–adrenal axis; TNFα: tumour necrosis factor-α; IL-1β: interleukin 1β; CCL-2: C-C motif chemokine ligand 2; CXCL-1: chemokine ligand 1. ↓ reduction; ↑ increase.

**Table 1 ijms-25-06473-t001:** Dietary interventions for the management of endometriosis-related pain.

Type of Dietary Intervention	Outcomes	Reference
Dietary protocols		
FODMAP diet	↓ pain	Moore et al., 2017 [248]
FODMAP diet	↓ *Akkermancia Muciniphila* ↓ *Actinobacteria* ↓ gut mucosal inflammation ↑ barrier function ↓ pain	Zhou et al., 2018 [250]
Low-fat vegetarian diet	↓ sex hormone binding protein ↓ pain intensity and duration ↓ premenstrual signs	Barnard et al., 2000 [252]
Mediterranean diet	↓ pain, dyspareunia, dysmenorrhea, dyschezia ↑ overall well-being	Ott et al., 2012 [257] Cirillo et al., 2023 [85]
Antioxidant diet	↑ antioxidant markers ↓ oxidative stress markers	Mier-Cabrera et al., 2009 [259]
Gluten-free diet	↓ inflammation ↓ pain	Marziali et al., 2012 [260]
Low-nickel diet	↓ pain, dyspareunia, dysmenorrhea ↓ GI symptoms ↓ extraintestinal symptoms	Borghini et al., 2020 [261]
Dietary supplementation		
*Lactobacillus gasseri*	↓ endometriosis progression and development ↓ pain	Itoh et al., 2011 [263]; Uchida et al., 2013 [264] Itoh et al., 2011 [265]
*Lactobacillus* sp.	↓ pain	Khodaverdi et al., 2019 [266]
Vitamins, mineral salts, VSL3 lactic ferments, omega-3, and omega-6	↓ pain ↑ quality of life	Sesti et al., 2007 [267]
Quercetin, curcumin, parthenium, nicotinamide, 5-methyltetrahydrofolate, and omega-3 and omega-6	↓ pain, dyspareunia, dysmenorrhea ↓ headache, muscle aches ↓ cystitis ↓ irritable colon	Signorile et al., 2018 [268]
Omega-3	↓ pain ↓ dysmenorrhea	Rahbar et al., 2012 [274]
Palmitoylethanolamine Transpolydatin	↓ pain	Cobellis et al., 2011 [276] Giugliano et al., 2013 [277] Indraccolo et al., 2010 [278]
Vitamin D	↓ pain ↓ PCR ↓ inflammation ↑ antioxidant capacity	Mehdizadehkashi et al., 2021 [280]
Vitamin D	No significant effects on pain	Almassinokiani et al., 2016 [281]

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
