# Peer review of "Endometriosis, Pain, and Related Psychological Disorders: Unveiling the Interplay among the Microbiome, Inflammation, and Oxidative Stress as a Common Thread"

_ijms, 2024, doi:10.3390/ijms25126473_

Round 1

Reviewer 1 Report

Comments and Suggestions for Authors

Dear Authors, thank you very much for this interesting paper. The topic is extremely relevant, and a comprehensive description of all these interrelationships enriches the existing literature.

I would like to mention some points:

(1) Please change the chapter number in line 720 and 875.

(2) You have carried out a very extensive literature research. Due to the large number of topixs, it is not possible to present the full literature analysis.But perhaps it is feasible to briefly name the matches and the exclusion criteria at the beginning of each chapter.

(3) I agree with your conclusion and future directions. Nevertheless, limitations with regard to the quality of the study results should be pointed out here (e.g. different study designs, small numbers of cases, lack of control groups, lack of prospektive studies). 

Author Response

  • Please change the chapter number in line 720 and 875.

Thank you for pointing out the incorrect chapter numbering. The correction has been made in the text.

  • You have carried out a very extensive literature research. Due to the large number of topics, it is not possible to present the full literature analysis. But perhaps it is feasible to briefly name the matches and the exclusion criteria at the beginning of each chapter.

Thank you for the interesting suggestion. As we analysed extensive literature for each chapter, considering the same types of studies (in vitro studies, animal models, observational clinical studies, and trials), we improved this classification and keywords once, at the end of the Introduction section. Regarding the exclusion criteria, we considered original articles and reviews published from 1994 until now, and we also considered only English-language publications.

  • I agree with your conclusion and future directions. Nevertheless, limitations with regard to the quality of the study results should be pointed out here (e.g. different study designs, small numbers of cases, lack of control groups, lack of prospective studies). 

According to the right suggestion of the reviewer, we included a more detailed discussion of the different limitations of current literature in the “conclusions” section.

Reviewer 2 Report

Comments and Suggestions for Authors

Dear Authors,

I would like to thank you for sharing this very interesting work and to applaud your efforts to elaborate this paper.

In my opinion, the article needs a bit of supplementary work, to rise to the journal‘s standards.

By streamlining the introduction, enhancing clarity in methods and results presentation, and providing personalized conclusions, the revised structure aims to engage the reader more effectively while addressing concerns about readability and content organization.

Patients and Methods

 - Description of the study population, including inclusion and exclusion criteria

- Data collection methods and outcome measures

- Flowchart depicting participant flow through the study

- Statistical analysis methods utilized

I believe the Results and Discussions need improvement.

- Interpretation of study findings in the context of existing literature

- Discussion of study limitations and potential biases

- Implications of the results for clinical practice and future research directions

Conclusion.

- Please specify the motivation behind this study and why you consider it a relevant and important topic for the gynecological community.

Best regards

Author Response

In my opinion, the article needs a bit of supplementary work, to rise to the journal‘s standards.

By streamlining the introduction, enhancing clarity in methods and results presentation, and providing personalized conclusions, the revised structure aims to engage the reader more effectively while addressing concerns about readability and content organization.

Patients and Methods

- Description of the study population, including inclusion and exclusion criteria

- Data collection methods and outcome measures

- Flowchart depicting participant flow through the study

- Statistical analysis methods utilized

Thank you for the insightful suggestions. Therefore, since this review analyses over 200 studies, we believe including all the requested information within each section would be overly verbose. An alternative could be to insert a table summarizing the characteristics of the analysed studies. However, as previously mentioned, given the large number of studies included, this could affect the manuscript readability .

I believe the Results and Discussions need improvement.

- Interpretation of study findings in the context of existing literature

- Discussion of study limitations and potential biases

- Implications of the results for clinical practice and future research directions

In agreement with the suggestion of the reviewer, we improved the interpretation of study findings within the framework of current literature and the limitations’ discussion at the conclusion paragraph's end. Furthermore, we added sentences addressing clinical implications and future research directions.

Conclusion.

- Please specify the motivation behind this study and why you consider it a relevant and important topic for the gynaecological community.

We specified the motivation behind this review at the end of the introduction. However, in line with the critical suggestion of the reviewer, we also emphasized this point at the end of the conclusions’ section.

Reviewer 3 Report

Comments and Suggestions for Authors

Nice review! Here are my comments:

Line 43: using “while” as a conjunction indicates a contrast, but these two sentences essentially have a similar interpretation.

Line 55: Is the “nevertheless” required to start this sentence?

It is better to also use the original references while referring to their results, in addition to another review article that you have referred now! For example, line 79 “ref21”, or line 52 “ref8”.

Line 115: I think “respect with” should be replaced by “with respect to”.

Line 117: regarding “Gambadauro et al.” report, this is a meta-analysis, not a report, so revise it. And provide the reference at the end of this sentence.

Line 120: revise this “by EM women, but without pain” for a better read.

Line 218: I think this sentence needs revision, to be more accurate in terms of the direct results from “ref75”.

Line 227-229: “ref78” does not have results regarding VEGEF role in endometriosis! So, revise with an appropriate reference.

Line 237: “ref81” result is irrelevant here!

Line 241: “ref83” is a review paper.

Line 245: provide the original reference for this data.

Line 252: I think there is no need for this many references here, probably only one recent reference noting this uncertainty would be good enough as a supporting reference.

Line 257: “ref63” is not a “recent” paper! So, replace by a recent one.

Line 259: “refs 92, 93” do these references discuss the role macrophages in EM?

Line 283: “ref102” is a review paper noting this finding, so add the original reference as well.

Line 294: “ref111” talks about macrophages and not mast cells. Is there any reference for mast cells?

Line 331: this sentence needs revision in terms of grammar.

Line 412: the “ref148” is irrelevant to this finding. And same for “ref63” in line 413. The next sentence also needs a reference.

Line 426: this finding needs a reference. Is it the “ref154”?

Line 580: provide the reference for this finding.

Line 603: add the original reference to this finding.

Line 672: provide the original reference.

Line 708: does that sentence “such as [237].” really end there?

Line 725: “ref1” avoids recommending nutrition to be effective, so you saying “supported” needs to be revised.

Line 809: “ref260” formatting is not complete.

Line 847: there is a tiny grammar error here, probably adding “and” will resolve it.

Author Response

Line 43: using “while” as a conjunction indicates a contrast, but these two sentences essentially have a similar interpretation.

Thank you for the right suggestion. We have rephrased the sentence in the text improving clarity.

Line 55: Is the “nevertheless” required to start this sentence?

It is better to also use the original references while referring to their results, in addition to another review article that you have referred now! For example, line 79 “ref21”, or line 52 “ref8”.

We have corrected the sentence in accordance with the suggestion. Furthermore, we have provided the references of the original works, replacing the previous ones that referred to review.

Line 115: I think “respect with” should be replaced by “with respect to”.

We have replaced the words as suggested.

Line 117: regarding “Gambadauro et al.” report, this is a meta-analysis, not a report, so revise it. And provide the reference at the end of this sentence.

We have revised the sentence to clarify the type of evidence being proposed according to the right suggestion.

Line 120: revise this “by EM women, but without pain” for a better read.

In agreement with the suggestion, we have revised the sentence.

Line 218: I think this sentence needs revision, to be more accurate in terms of the direct results from “ref75”.

We have rephrased the remarked sentence to better clarity the results of the referenced study.

Line 227-229: “ref78” does not have results regarding VEGEF role in endometriosis! So, revise with an appropriate reference.

We sorry for the mistake,  we added the appropriate reference (Park et al.).

Line 237: “ref81” result is irrelevant here!

Thank you for bringing it to our attention; we have removed this reference.

Line 241: “ref83” is a review paper.

We have included original references in the text.

Line 245: provide the original reference for this data.

We have provided the correct reference in the text.

Line 252: I think there is no need for this many references here, probably only one recent reference noting this uncertainty would be good enough as a supporting reference.

Thank you for your insightful suggestion; we have removed the redundant references adding a recent and comprehensive reference (Cacciottola et al.) regarding this topic.

Line 257: “ref63” is not a “recent” paper! So, replace by a recent one.

We have replaced Asante et al. (2011) with Gharaei et al. (2023).

Line 259: “refs 92, 93” do these references discuss the role macrophages in EM?

Thank you for the constrictive suggestion. We have revised the text to better  explain the role of macrophages and endothelial cells on the production of VEGF and NGF during endometriosis.

Line 283: “ref102” is a review paper noting this finding, so add the original reference as well.

Thank you for the suggestion. The sentence refers to the original study by Tokushige et al. and not to a review, as indicated in the comment. The original work investigates how hormonal treatment can change nerve fiber density and aims to identify types of nerve fibers in the endometrium and myometrium in endometriosis patients.

Line 294: “ref111” talks about macrophages and not mast cells. Is there any reference for mast cells?

Thank you for specifying this aspect; we have provided appropriate references to support this sentence.

Line 331: this sentence needs revision in terms of grammar.

Thank you for pointing out the error; we have correctly rewritten the sentence.

Line 412: the “ref148” is irrelevant to this finding. And same for “ref63” in line 413. The next sentence also needs a reference.

We have adjusted the references as suggested.

Line 426: this finding needs a reference. Is it the “ref154”?

We have added the correct reference in the text.

Line 580: provide the reference for this finding.

We have provided the reference.

Line 603: add the original reference to this finding.

We have added the accurate reference in the text.

Line 672: provide the original reference.

We have added the accurate reference.

Line 708: does that sentence “such as [237].” really end there?

We have corrected the remarked mistake.

Line 725: “ref1” avoids recommending nutrition to be effective, so you saying “supported” needs to be revised.

We have reviewed and rewritten this sentence for greater clarity.

Line 809: “ref260” formatting is not complete.

We have correctly formatted the reference.

Line 847: there is a tiny grammar error here, probably adding “and” will resolve it.

We have corrected the grammar error by adding "and" as suggested.
